# Nanophotonics for pair production

**Valerio Di Giulio** [1] **& F. Javier García de Abajo** [1,2] ✉

The transformation of electromagnetic energy into matter represents a fascinating prediction of relativistic quantum electrodynamics that is paradigmatically exemplified by the creation of electron-positron pairs out of light. However, this phenomenon has a very low probability, so positron sources rely instead on beta decay, which demands elaborate monochromatization and trapping schemes to achieve high-quality beams. Here, we propose to use intense, strongly confined optical near fields supported by a nanostructured material in combination with high-energy photons to create electron-positron pairs. Specifically, we show that the interaction between near-threshold $\gamma$-rays and polaritons yields higher pair-production cross sections, largely exceeding those associated with free-space photons. Our work opens an unexplored avenue toward generating tunable pulsed positrons from nanoscale regions at the intersection between particle physics and nanophotonics.

The creation of massive particles from electromagnetic energy emerged as a prominent focus of attention in 1934, when the materialization of an electron and its antiparticle−the positron−was predicted to occur with nonvanishing probability by Breit and Wheeler (BW) from the scattering of two photons[1], by Bethe and Heitler (BH) from the interaction of a photon and the Coulomb potential of a nucleus[2], and by Landau and Lifshitz (LL) from the collision of two other massive particles[3]. A main difference between these processes relates to the real or virtual nature of the involved photons. While only real electromagnetic quanta lying inside the light cone (i.e., satisfying the light dispersion relation in vacuum, $k = \omega/c$) participate in the BW mechanism for pair production, the LL process is mediated by two virtual photons, and both real and virtual photons play a role in BH scattering. Eventually, pair production was achieved by colliding energetic electrons and real photons delivered by high-power lasers[4], and more recently using only real photons generated from atomic collisions[5].

Besides the fundamental interest of these processes, the generation of positrons finds application in surface science[6] through, for example, positron annihilation spectroscopy[7–9] and low-energy positron diffraction[10], as well as in the study of their interactions with atoms and molecules[11,12]. Positrons are also used to create antimatter, for example, antihydrogen[13–16] and positronium[17]. In these studies, slow positrons are commonly obtained from beta decay, decelerated through metallic moderators[18], and subsequently stored in different types of traps, from which they are extracted as low-energy, quasi-monochromatic pulses[19–22].

Direct positron generation from light would not require nuclear decay and could further leverage recent advances in optics to produce ultrashort photon pulses. However, the cross-sections associated with the aforementioned processes are extremely small. As a possible avenue to increase the pair-production rate, we consider the replacement of free photons by confined optical modes in the hope that they alleviate the kinematic mismatch between the particles involved in BW scattering. In particular, surface polaritons, which are hybrids of light and polarization charges bound to material interfaces, can display short in-plane wavelengths compared with the free-space light wavelength. Such modes involve electromagnetic energy trapped at the interface between two media with different dielectric properties. For example, for a planar interface, when the sign of the real part of the permittivity of the two media is opposite, the associated optical fields decay exponentially away from the interface, but a similar behavior is observed when light is trapped by a thin metallic film[23]. Likewise, light can be trapped in polaritons sustained by more involved geometries[24] (e.g., in a self-standing sphere, a polariton is defined by the condition that its permittivity is equal to −2 if one neglects retardation effects). Actually, a broad suite of two-dimensional (2D) materials has recently been identified to sustain long-lived, strongly confined polaritons[25,26], including plasmonic[23,27,28], phononic[29,30], and excitonic[31] modes that cover a wide spectral range extending from mid-infrared frequencies[23,27,29,30] to the visible domain[28,31]. Specifically, modes bound to nanogaps[32] feature large field confinement and enhancement (in vacuum

[1]ICFO-Institut de Ciencies Fotoniques, The Barcelona Institute of Science and Technology, 08860 Castelldefels, Barcelona, Spain. [2]ICREA-Institució Catalana de Recerca i Estudis Avançats, Passeig Lluís Companys 23, 08010 Barcelona, Spain. ✉e-mail: javier.garciadeabajo@nanophotonics.es

regions) that boost light-mediated processes, such as surface-enhanced Raman scattering (SERS).

In this work, we calculate the pair-production cross section associated with the annihilation of $\gamma$-ray photons ($\gamma$-photons) and confined polaritons, leading to a substantial enhancement compared to free-space BW scattering. Part of this enhancement relates to the spatial confinement of surface polaritons, as the lack of translational invariance enables pair production for $\gamma$-photon energies just above the $2m_e c^2$ threshold (e.g., at the $^{60}$Co emission line $\hbar\omega_\gamma$ -1.17 MeV combined with a polariton energy $\hbar\omega_p$ of a few eV), in contrast to free-space BW scattering, for which visible-range photons need to be paired with GeV photons such as those generated in astrophysical processes[33,34]. The latter include absorption of high-energy $\gamma$ rays by extra-galactic background light[35], by active galactic nuclei[36], and during $\gamma$-ray bursts[37], as well as plasma production in neutron-star magnetospheres[38]. For polaritonic nanogap modes confined in three dimensions, pairs can be produced by $\gamma$-photon scattering in the gap vacuum region, where polariton-mediated positron emission is not affected by the background of other emission processes such as BH scattering. By demonstrating the advantages of using deeply confined light, our work inaugurates an avenue in the exploration of nanophotonic structures as a platform for high-energy physics.

## Results

### Pair production from the scattering of a polariton and a $\gamma$-photon

Considering the general configuration illustrated in Fig. 1a, we study pair production by using the relativistic minimal coupling Hamiltonian[39,40]

$$\hat{\mathcal{H}}_{\text{int}}(t) = \frac{-1}{c} \int d^3\mathbf{r}\, \hat{\mathbf{j}}(\mathbf{r}) \cdot \mathbf{A}(\mathbf{r}, t), \tag{1}$$

where $\hat{\mathbf{j}}(\mathbf{r}) = -ec : \overline{\Psi}(\mathbf{r}) \overrightarrow{\gamma} \hat{\Psi}(\mathbf{r}) :$ is the fermionic current, $\mathbf{A}(\mathbf{r}, t)$ is the classical vector potential associated with the polariton and photon fields, and we adopt a gauge with vanishing scalar potential. Here, $: \cdot :$ denotes normal product concerning electron and positron

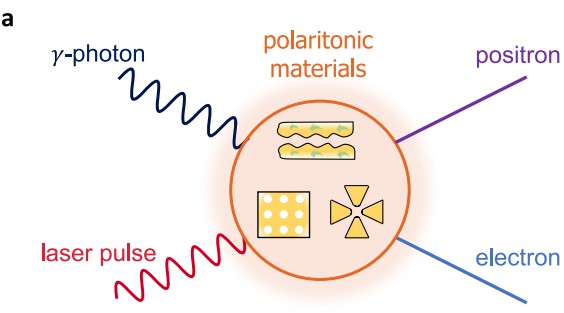

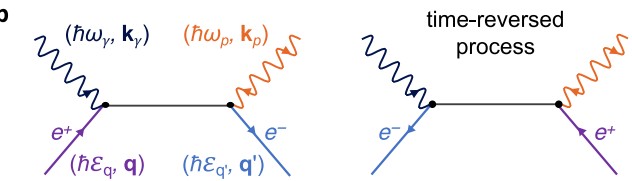

**Fig. 1 | Pair production by interaction of polaritons and $\gamma$-photons. a** We consider polaritons supported by a material structure. Energetic $\gamma$-rays interact with the polaritons, giving rise to electron–positron pairs. **b** Direct and time-reversed Feynman diagrams contributing to the investigated pair production. We indicate the energies and wave vectors of the polariton, the $\gamma$-photon, and the fermions by color-coordinated labels. Both polariton emission and absorption processes contribute to pair production, as indicated by orange arrows pointing toward the vertex or away from it, respectively.

annihilation ($\hat{c}_{\mathbf{q},s}$ and $\hat{d}_{\mathbf{q},s}$, respectively) and creation ($\hat{c}^\dagger_{\mathbf{q},s}$ and $\hat{d}^\dagger_{\mathbf{q},s}$) operators (for fermions of momentum $\hbar\mathbf{q}$, spin $s$, and energy $\hbar\varepsilon_q = c\sqrt{m_e^2 c^2 + \hbar^2 q^2}$), and $\hat{\Psi}(\mathbf{r}) = \sum_{\mathbf{q},s}(u_{\mathbf{q},s}\hat{c}_{\mathbf{q},s}e^{i\mathbf{q}\cdot\mathbf{r}} + v_{\mathbf{q},s}\hat{d}^\dagger_{\mathbf{q},s}e^{-i\mathbf{q}\cdot\mathbf{r}})$ is the fermionic field operator, with $u_{\mathbf{q},s}$ ($v_{\mathbf{q},s}$) representing 4-component electron (positron) spinors.

We work in the continuous-wave regime and eventually normalize the resulting production rate to the number of polaritons and photons in the system. The vector potential is thus $\mathbf{A}(\mathbf{r}, t) = -(ic/\omega_p) \overrightarrow{\mathcal{E}}_p(\mathbf{r})e^{-i\omega_p t} - (ic/\omega_\gamma) \mathcal{E}_\gamma \hat{\mathbf{e}}_j e^{ik_\gamma z - i\omega_\gamma t} + \text{c.c.}$ (i.e., the sum of two monochromatic components), in which we consider two different polarizations $\hat{\mathbf{e}}_j = \hat{\mathbf{x}}$ or $\hat{\mathbf{y}}$ (with $j = 1$ or 2) for the $\gamma$-ray field and take it to propagate along the $z$ direction with the wave vector $\mathbf{k}_\gamma = \hat{\mathbf{z}}\,\omega_\gamma/c$. The polaritonic electric field $\mathbf{E}(\mathbf{r}, t) = \overrightarrow{\mathcal{E}}_p(\mathbf{r})e^{-i\omega_p t} + \text{c.c.}$ incorporates a spatially localized amplitude $\overrightarrow{\mathcal{E}}_p(\mathbf{r})$, for which we use expressions that describe modes in planar surfaces[23] and nanostructures[24]. Such expressions have been extensively and successfully used in the explanation of nanophotonics experiments[23,24,41].

The production rate for a state $\hat{d}^\dagger_{\mathbf{q},s}\hat{c}^\dagger_{\mathbf{q}',s'}|0\rangle$ comprising a positron (wave vector $\mathbf{q}$, spin $s$) and an electron (wave vector $\mathbf{q}'$, spin $s'$) is then calculated to the lowest (second) nonvanishing-order of time-dependent perturbation theory for the Hamiltonian in Eq. (1). This level of perturbation should be sufficient considering the low obtained cross sections (see below), while the renormalization group[42,43] could be used to account for nonperturbative corrections. Following a standard procedure detailed in Supplementary Notes 1 and 2, the positron-momentum-resolved pair-production cross section associated with polariton and $\gamma$-photon scattering is found to be

$$\frac{d\sigma^{\text{pol}}}{d\mathbf{q}} = \frac{\alpha^2 c^5}{32\pi^4 N_p \hbar\omega_\gamma \omega_p^2} \int d^3\mathbf{q}' \sum_{\pm} \sum_{ss'j}$$
$$\delta(\varepsilon_q + \varepsilon_{q'} - \omega_\gamma \pm \omega_p)\left|\overline{u}_{\mathbf{q}'s'}\mathcal{M}_j^\pm(\mathbf{q}', \mathbf{q})v_{\mathbf{q}s}\right|^2, \tag{2a}$$

where $\alpha \approx 1/137$ is the fine-structure constant, we average over $\gamma$-ray polarizations $j$, and a $4 \times 4$ matrix

$$\mathcal{M}_j^\pm(\mathbf{q}', \mathbf{q}) = \gamma^j\, G_F(\mathbf{q}' - \mathbf{k}_\gamma, \varepsilon_{q'} - \omega_\gamma)\,\overrightarrow{\gamma} \cdot \overrightarrow{\mathcal{E}}^\pm_{p, \mathbf{q}+\mathbf{q}'-\mathbf{k}_\gamma}$$
$$+ \overrightarrow{\mathcal{E}}^\pm_{p, \mathbf{q}+\mathbf{q}'-\mathbf{k}_\gamma} \cdot \overrightarrow{\gamma}\, G_F(\mathbf{k}_\gamma - \mathbf{q}, \omega_\gamma - \varepsilon_q)\,\gamma^j \tag{2b}$$

is defined in terms of the Dirac $\gamma$ matrices, the Feynman propagator[40] $G_F(\mathbf{q}, \omega) = [\omega\gamma^0 - c\overrightarrow{\gamma} \cdot \mathbf{q} + (m_e c^2/\hbar)]/(\omega^2 - \varepsilon_q^2 + i0^+)$, and the momentum representation of the polariton field $\overrightarrow{\mathcal{E}}_{p, \mathbf{k}_p} = \int d^3\mathbf{r}\, \overrightarrow{\mathcal{E}}_p(\mathbf{r})e^{-i\mathbf{k}_p \cdot \mathbf{r}}$ and $\overrightarrow{\mathcal{E}}^+_{p, \mathbf{k}_p} = (\overrightarrow{\mathcal{E}}^-_{p, \mathbf{k}_p})^*$. The cross-section in Eq. (2a) is normalized per polariton and incident $\gamma$-photon, and in particular, the denominator in front of the integral contains the number of polaritons $N_p$ sustained by the $\overrightarrow{\mathcal{E}}_p(\mathbf{r})$ field (see Supplementary Notes 1 and 2 for details).

Equations (2a) and (2b) describes the annihilation of a $\gamma$-photon accompanied by the emission (upper signs) or absorption (lower signs) of a polariton, as indicated in the Feynman diagrams in Fig. 1b (orange arrows pointing away or toward the vertex, respectively), where a finite range of wave vectors $\mathbf{k}_p$ is generally involved due to spatial confinement. Incidentally, we note that boson emission is forbidden in free-space BW scattering, whereas it contributes to the present polariton-mediated pair-production process. In addition, polariton absorption and emission processes lead to nearly identical contributions to the pair-production cross-section because the polariton energy and momentum are small compared with the total transferred amounts of these quantities.

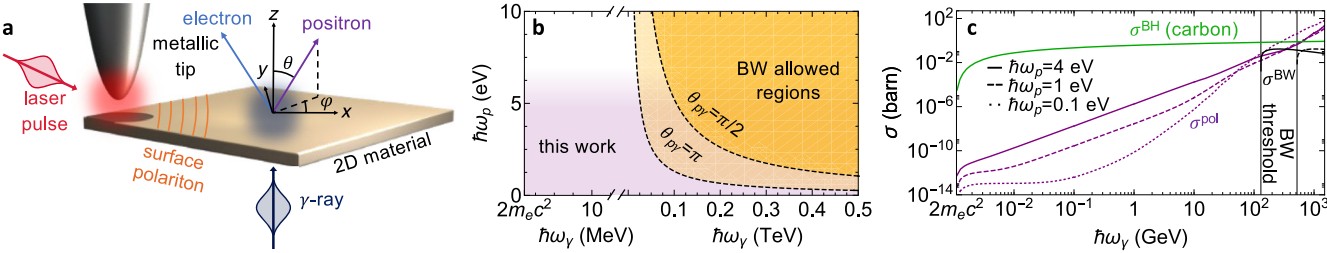

**Fig. 2 | Pair-production assisted by surface polaritons. a** We consider surface modes excited in a 2D material by a coupling tip illuminated by laser pulses (red), while the γ-rays (dark gray) normally impinge on the surface. The positron emission direction $(\theta, \varphi)$ (purple) determines the electron direction (blue) by conservation of energy and in-plane momentum. **b** Comparison between the regions allowed by energy–momentum conservation in either Bethe–Wheeler (BW) photon–photon scattering (yellow) and polariton–photon scattering under the configuration of Fig. 1a (purple) as a function of polariton/photon energies. The BW threshold

$\hbar^2 \omega_p \omega_\gamma = 2m_e^2 c^4/(1 - \cos\theta_{p\gamma})$[39] is indicated for a relative photon-photon angle $\theta_{p\gamma}$ of $\pi$ (absolute threshold) and $\pi/2$. **c** Pair-production cross sections for polariton-photon scattering ($\sigma^{\rm pol}$, purple curves), BW scattering ($\sigma^{\rm BW}$ for $\theta_{p\gamma} = \pi/2$, black curves; see Supplementary Note 5.3), and Bethe–Heitler (BH) scattering by a carbon atom ($\sigma^{\rm BH}$, green curve; see Supplementary Note 5.1). We consider different polariton energies (see legend) with a fixed $k_p = 0.05$ nm$^{-1}$ in all cases. Solid vertical lines indicate the γ-photon BW threshold energies taken from the $\theta_{p\gamma} = \pi/2$ curve in (**a**).

## Pair production assisted by surface polaritons

As an illustrative scenario, we consider surface polaritons (frequency $\omega_p$, wave vector $\mathbf{k}_p = k_p \hat{\mathbf{x}}$) launched on a 2D material ($z = 0$ plane) by in-coupling a laser through a metallic tip (or, alternatively, a grating) (see Fig. 2a), producing a polariton field amplitude that can be generally written as (see, for example, ref. 44) $\vec{\mathcal{E}}_p(\mathbf{r}) \propto [ik_p\hat{\mathbf{x}} - k_p\mathrm{sign}\{z\}\hat{\mathbf{z}}] e^{ik_p x - \kappa_p|z|}$ with $\kappa_p = \sqrt{k_p^2 - \omega_p^2/c^2}$, where we neglect material losses, γ-ray screening, and finite-thickness effects. Parallel momentum conservation leads to $\mathbf{q}'_{\parallel\pm} = -\mathbf{q}_\parallel \mp \mathbf{k}_p$ for the in-plane electron wave vector components, while energy conservation determines the electron energy $\varepsilon_{q'_\pm} = \omega_\gamma \mp \omega_p - \varepsilon_q$ and out-of-plane wave vector $q'_{z\pm} = \sqrt{\varepsilon_{q'_\pm}^2/c^2 - m_e^2 c^2/\hbar^2 - q'^2_{\parallel\pm}}$, subject to the threshold-energy conditions $\varepsilon_{q'_\pm}^2 > m_e^2 c^4/\hbar^2 - c^2 q'^2_{\parallel\pm}$ and $\omega_\gamma > \pm\omega_p + \varepsilon_q$. Calculating the Fourier transform of the surface polariton field and inserting it into Eqs. (2), we find (see Supplementary Note 4)

$$\frac{d\sigma^{\rm pol}}{d\mathbf{q}} = \frac{\alpha^2 c^3 \kappa_p}{\pi\,\omega_p\omega_\gamma k_p^2} \sum_\pm \frac{\varepsilon_{q'_\pm}}{q'_{z\pm}}$$
$$\times \sum_{ss'j\mu} \left| \bar{u}_{\mathbf{q}'_{\mu\pm}, s'} \mathcal{N}_j^\pm(\mathbf{q}'_{\mu\pm}, \mathbf{q}) v_{\mathbf{q}s} \right|^2, \tag{3}$$

where $\mathbf{q}'_{\mu\pm} = \mathbf{q}'_{\parallel\pm} + \mu q'_{z\pm}\hat{\mathbf{z}}$ is the electron wave vector for upward ($\mu = 1$) and downward ($\mu = -1$) emission contributions, while $\mathcal{N}_j^\pm(\mathbf{q}', \mathbf{q})$ is given by Eq. (2b) with $\vec{\mathcal{E}}_{p,\mathbf{q}+\mathbf{q}'-\mathbf{k}_p}$ replaced by a vector $\mathbf{f}_{\pm(k_{yz}-q_z-q'_z)}$ defined in such a way that $\mathbf{f}_{k_z} = (\kappa_p^2\hat{\mathbf{x}} + k_p k_z\hat{\mathbf{z}})/(\kappa_p^2 + k_z^2)$ encapsulates the out-of-plane momentum distribution of the polariton field.

An immediate consequence of out-of-plane symmetry breaking is that the allowed kinematical space for which we obtain nonzero pair-production cross sections extends down to the infrared polariton regime even when using γ-photons just above the absolute energy threshold $\gtrsim 2m_e c^2 \approx 1.02$ MeV (Fig. 2b). In contrast, BW scattering with one of the photons in the optical regime requires the other photon to have energy exceeding ~0.1 TeV, which explains why free-space pair production has traditionally been observed only in its nonlinear version, where the energy–momentum mismatch is overcome by engaging a high number of photon exchanges[45,46].

In Fig. 2c, we show that, for low-energy polaritons/photons (up to a few eV), the momentum-integrated polariton-assisted pair-production cross-section $\sigma^{\rm pol} = \int d^3\mathbf{q}\,(d\sigma^{\rm pol}/d\mathbf{q})$, with $d\sigma^{\rm pol}/d\mathbf{q}$ given by Eq. (3), takes substantial values at γ-photon energies far below the BW kinematical threshold (vertical solid lines). In addition, $\sigma^{\rm pol}$ is consistently several orders of magnitude higher than the BW cross section

up to γ-photon energies in the TeV regime. Part of this enhancement can be attributed to the effect of spatial compression of polaritons relative to free-space photons.

Upon numerical examination of Eq. (3), we find positron emission to be dominated by contributions associated with an equal partition of kinetic energy between the two fermions, both for near-threshold (Supplementary Fig. 2) and GeV (Supplementary Fig. 3) emission, also displaying sharp angular profiles peaked around the forward direction defined by the γ-ray.

Unfortunately, the emission arising from scattering by surface polaritons is orders of magnitude smaller than that associated with BH scattering by the polaritonic material, as revealed by comparing their respective cross sections normalized per polariton and per atom (Fig. 3c). For example, for 1.17 MeV γ-photons traversing a highly doped graphene monolayer that supports 1 eV plasmons, the ratio between the emission from these two mechanisms is $(n_p\sigma^{\rm pol})/(n_C\sigma^{\rm BH})$, where $n_p$ is the plasmon surface density, $n_C \sim 40/{\rm nm}^2$ is the carbon atom density, and we have $\sigma^{\rm pol} \sim 10^{-13}$ barn (1 barn $= 10^{-24}$ cm$^2$) and $\sigma^{\rm BH} \sim 10^{-4}$ barn[47] (see Supplementary Note 5.1). For the two signals to be comparable in magnitude, an unrealistically large plasmon density $n_p > 10^{10}/{\rm nm}^2$ would be required.

## Threshold pair-production assisted by gap polaritons

To reduce the effect of the BH background, we study pair production by scattering of γ-photons and gap polaritons (Fig. 3a). Besides the emission enhancement expected from the breaking of translational invariance in all directions, positrons produced by gap polaritons and γ-photons arise from the vacuum gap region, where no BH signal is generated, thus facilitating the identification of a polariton-assisted pair-production signal (see further discussion below). For simplicity, we consider a polariton field described by $\vec{\mathcal{E}}_p(\mathbf{r}) = E_p\hat{\mathbf{x}}\,\Theta(R_p - r)$, which has a uniform amplitude $E_p$ polarized along $x$ and extending within a sphere of radius $R_p$. This simple expression defines a confined mode of size $R_p$ that allows us to obtain analytical expressions, while the details of the field in more complex structures[24] should only produce minor modifications in the final results. Inserting this field in Eqs. (2a) and (2b), we obtain (see a detailed derivation in Supplementary Note 3)

$$\frac{d\sigma^{\rm pol}}{d\mathbf{q}} = \frac{3\alpha^2 c^3}{32\pi^4\omega_\gamma\omega_p R_p^3} \int d\Omega_{\mathbf{q}'} \sum_\pm q'_\pm \varepsilon_{q'_\pm}$$
$$\times \sum_{ss'} \sum_{j=1,2} \left| \bar{u}_{\mathbf{q}'_\pm, s'} \mathcal{P}_j^\pm(\mathbf{q}'_\pm, \mathbf{q}) v_{\mathbf{q}s} \right|^2, \tag{4}$$

**a**

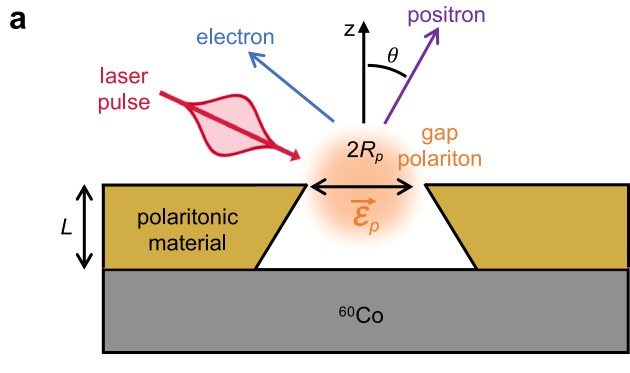

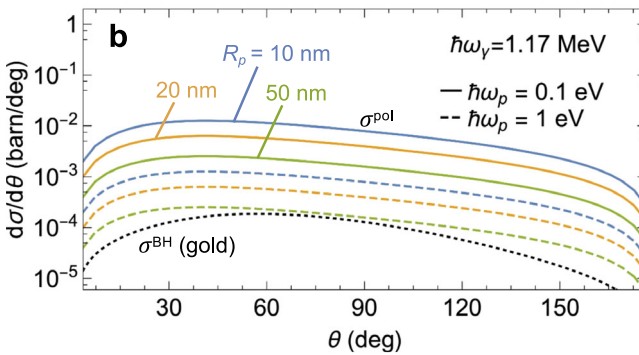

**Fig. 3 | Pair-production from a gap polariton. a** Sketch of the geometry under consideration, in which pairs are produced by $\gamma$-photons traversing a gap polariton. The latter can be excited by a laser pulse and is taken to have frequency $\omega_p$ and uniform field $\vec{\mathcal{E}}_p$ confined to a spherical region of radius $R_p$ (flanked by a polaritonic material). **b** Differential cross-section as a function of polar angle for polariton-assisted positron emission under the configuration in (**a**) (colored curves for different values of $R_p$ and $\omega_p$, as indicated by labels), compared with the BH cross section for a gold atom (see Supplementary Note 5.1). We consider 1.17 MeV $\gamma$-photons in all cases.

where the value of $q'_\pm$ is determined by the energy conservation condition $\varepsilon_{q'_\pm} = \omega_\gamma - \varepsilon_q \mp \omega_p$, the integral is restricted by the condition $\omega_\gamma \mp \omega_p \geq m_e c^2/\hbar + \varepsilon_q$, and we define $\mathcal{P}_j^\pm(\mathbf{q}', \mathbf{q}) = \mathcal{M}_j^\pm(\mathbf{q}', \mathbf{q})/E_p$ [see Eq. (2b)]. The latter involves the normalized Fourier transform of the polariton field amplitude $\vec{\mathcal{E}}_p(\mathbf{r})$, which reduces to

$$\frac{\vec{\mathcal{E}}_{p,\mathbf{k}}}{E_p} = \frac{4\pi \hat{\mathbf{x}}}{k^3}\left[\sin(kR_p) - kR_p\cos(kR_p)\right].$$

We use Eq. (4) to compute the results presented in Fig. 3b for different polariton sizes $R_p$ and energies $\hbar\omega_p$ after integrating over the azimuthal angle of positron emission. The differential cross section normalized per polariton and $\gamma$-photon exhibits a monotonic increase with decreasing $R_p$ and $\omega_p$ as well as a smooth dependence on polar angle $\theta$.

Once more, we need to compare polariton-driven pair production to the background BH positron signal (i.e., $\gamma$-ray scattering by the nuclei of the polaritonic material). The complete suppression of BH scattering from the vacuum gap region could be leveraged by selecting positrons originating only in that region through the use of charged-particle optics elements (i.e., a positron analog of electron optics in an electron microscope), such that only positrons coming from the gap are collected, similarly to how photoemission electron microscopes collect electrons emitted within specimen regions spanning just a few nanometers[48].

Even without resorting to positron microscopy, we argue next that spatial confinement in gap polaritons leads to a discernible positron emission signal under laser pulse irradiation when compared to the BH background, as the cross section per polariton undergoes an

increase by several orders of magnitude when moving from confinement in one direction (surface polaritons, Fig. 2c) to full three-dimensional trapping (gap polaritons, Fig. 3b). For concreteness, we focus on low-energy ($\hbar\omega_p = 0.1$ eV) gap plasmons confined to an opening in a gold film with an effective mode volume assimilated to a sphere of radius $R_p = 50$ nm. These parameters can be obtained by engineering the morphology of the metal gap[32]. In practice, we envision an array of gaps such that the openings span a fraction $\eta$ of the film surface. Under illumination with a laser peak amplitude of $10^8$ V/m (a typical value below the damage threshold when using ultrafast pulses[49]) and a realistic polaritonic field enhancement of $10^2$ (i.e., $E_p \sim 10^{10}$ V/m), we have a number of polaritons $N_p \approx E_p^2 R_p^3/3\hbar\omega_p \sim 3\times 10^7$ per gap (i.e., a surface polariton density $n_p = \eta N_p/\pi R_p^2 \sim 4\,\eta\times 10^3/\text{nm}^2$; see Supplementary Note 4), and therefore, the fraction of positrons generated per $\gamma$-photon is $n_p\sigma^{\text{pol}} \sim \eta\times 10^{-7}$, where we take $\sigma^{\text{pol}} \sim 0.25$ barn for the pair-production cross section per polariton (see Fig. 3b).

This fraction has to be compared to that of positrons associated with the BH mechanism. For a gold film of thickness $L = 100$ nm (much smaller than the positron escape depth; see Supplementary Note 5.2), as commonly employed in plasmonic studies, we combine the BH cross section for a gold atom at 1.17 MeV $\gamma$-photon energy ($\sigma^{\text{BH}} \approx 16$ mbarn; see Fig. 3b) together with the volume per gold atom $\mathcal{V} \approx 17.0$ Å$^3$ (i.e., a surface gold atom density $n_{\text{Au}} = L/\mathcal{V} \sim 6\times 10^3/\text{nm}^2$), to compute the fraction of BH positrons per incident photon, $n_{\text{Au}}\sigma^{\text{BH}} \sim 10^{-8}$. Under these conditions, the ratio of polariton-assisted emission to BH emission is $\sim 10\,\eta$. For a realistic value of the opening fraction $\eta \sim 10\%$, the noted ratio becomes $\sim 1$, and therefore, polariton-mediated pair production and BH scattering are comparable in magnitude.

We remark that this estimate assumes a synchronized detection, such that the signal is only collected within the duration of the optical pulses needed to sustain a large number of polaritons in the system. For example, with 1 g of $^{60}$Co, we have $\sim 100$ $\gamma$-photons overlapping in time with the duration of a 1 ps laser pulse (see Supplementary Note 4), which leads to the emission of $n_p\sigma^{\text{pol}} \sim 10^{-6}$ positrons per pulse, half of them produced by polariton-assisted scattering. We thus predict a measurable signal when employing a high-repetition ($\sim 10^8$ Hz) pulsed laser.

As an alternative geometry, one could rely on polaritons confined to nanoparticles (e.g., gold colloids[24]) of similar size as the gaps considered above and dispersed on a thin film (e.g., monolayer graphene), leading to similar estimates for the positron production yield and even higher ratios of polariton-driven to BH positron emission because of the reduction in polaritonic material volume.

## Discussion

In conclusion, we advocate for the use of optical excitations confined to nanostructured materials in combination with $\gamma$-rays as a way of producing electron-positron pairs with higher efficiency than free-space BW scattering and requiring substantially lower photon energies. The breaking of translational invariance is responsible for the latter, whereas the spatial compression of the optical fields associated with surface polaritons facilitates the coupling to high-momentum products (the fermions). The proposed mechanism is still orders of magnitude weaker than BH scattering when the pairs are produced by $\gamma$-photons traversing polariton-supporting materials (e.g., graphene and planar waveguides). Isolation of the proposed mechanism could be achieved with the use of focused $\gamma$-photons targeted at the region in which the polariton has high strength, although such focusing represents a pending challenge in itself. Alternatively, we argue that gap polaritons confined to vacuum regions flanked by such materials can circumvent this problem by, for example, collecting positrons created at the gap using a positron microscope. In addition, when synchronizing positron detection with exposure of gap polaritons to ultrafast laser pulses, we show that the BH background becomes comparatively small thanks to the enhancement in the emission associated with strong polariton spatial confinement in

all three dimensions. We remark that these conclusions are drawn from the study of positron emission produced by near-threshold $\gamma$-photons, such as those from $^{60}$Co.

Besides its fundamental interest, the proposed mechanism for polariton-driven positron emission opens exciting possibilities that are not accessible to other types of positron sources, such as the generation of positron pulses with ultrafast durations inherited from the incident laser pulses, as well as the nanoscale size of those sources when relying on confined gap polaritons. We envision the spatiotemporal modulation of the positron wave functions by shaping the employed laser field or by tailoring the spatial distribution and polarization of the polaritonic field (e.g., to create chiral positron beams or two-pulse positron states). We remark that the localized nature of the emission from regions in which strongly confined and intensity-enhanced polaritons are sustained renders this mechanism appealing for applications that demand spatially confined positron sources. Rough metal surfaces should also boost pair production due to the large optical enhancement that takes place at plasmonic hotspots[32] in analogy to SERS, although positron emission via the BH mechanism would result in a large signal in such structures (see discussion on generation from gold films above) that could be reduced by resorting to ultrathin self-standing films or, alternatively, by synchronizing positron detection and laser pulse excitation of the polaritonic field. The inverse process of positron annihilation stimulated by polaritonic fields upon impact on a polariton-supporting material also bears interest as a possible source of localized $\gamma$-photons. Beyond these potential uses, antimatter production and annihilation assisted by collective optical excitations bear intrinsic interest as an example of a nanophotonics approach to high-energy physics.

## Data availability

The data that support the findings of this study are available from the corresponding author upon request.

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

## Acknowledgements

This work has been supported in part by the European Research Council (Advanced Grant 789104-eNANO), the Spanish MICINN (PID2020-112625GB-I00 and Severo Ochoa CEX2019-000910-S), the Catalan CERCA Program, and Fundacións Cellex and Mir-Puig.

## Author contributions

F.J.G.A. conceived the concept. V.D.G. and F.J.G.A. developed the theory and wrote the paper. V.D.G. performed the calculations and prepared the figures.

## Competing interests

The authors declare no competing interests.
