## [Peer Review File · Nature Communications]

REVIEWER COMMENTS

Reviewer #1 (Remarks to the Author):

This paper proposes an interesting mechanism in order to produce electron-positron pairs based on the interaction between the electromagnetic field generated in polaritonic modes and a gamma ray. The calculation of the corresponding interaction cross-section is carried out following standard procedures, and the paper comes with pedagogical appendices on calculation techniques which will certainly prove useful on their own. The main result of this calculation is that the cross-section can be much larger than the Bethe-Heitler cross-section for gap polaritons, thus opening new venues for pair production.

The paper is well written and clearly structured. I have only a few comments.

1) In introduction pair production mechanisms and experimental realisations are reviewed. One of the main points being raised is that the $gg \rightarrow e^+e^-$ (Breit-Wheeler) scattering has too low a cross-section to be practical. As a suggestion, I think the paper could benefit from a broadening of the scope of the introduction by mentioning the important astrophysical implications of this process. A general review was done in <https://ui.adsabs.harvard.edu/abs/2010PhR...487....1R/abstract>. Some more specific examples with suggested references are listed below:

- absorption of TeV gamma rays by the extra-galactic background light
<https://ui.adsabs.harvard.edu/abs/2015ApJ...812...60B/abstract>

- absorption of high-energy gamma rays in the immediate environment of compact sources such as Active Galactic Nuclei (AGN) or during gamma-ray bursts (GRBs)
<https://ui.adsabs.harvard.edu/abs/2019A%26A...627A.159H/abstract> ,
<https://academic.oup.com/mnras/article/475/1/L1/4693845>

- plasma production in neutron-star magnetospheres
<https://ui.adsabs.harvard.edu/abs/2018MNRAS.474.1436V/abstract>

2) Not being a specialist of polaritons, I cannot evaluate in detail the relevance of all the statements made concerning these physical objects. Given that the paper is published in a journal targeted at a broad audience, I would recommend to give more references on general polariton physics and in particular the expressions used for their electric field since this is of prime importance for the obtained results.

Reviewer #2 (Remarks to the Author):

This paper proposes an entirely novel method of producing positrons from the collision of two photons to create an electron/positron pair. The novelty arises in that one of the photons has relative low energy of the order of 1eV in contrast to the second member of the pair which would be a gamma ray. Cross sections for this process would normally be too small to be of use but in this work enhancement is sought by capturing the low energy photons in a polaronic system where the intensity of the polaronic field can be concentrated in a narrow gap. This concentration is a well understood process but to my knowledge has not been exploited in the context of positron production. According to the authors' calculations the method gives several orders of magnitude improvement over existing technology.

The authors might consider whether a simpler solution would be a rough silver surface to enhance the polaronic fields, as in giant Raman scattering.

Derivation of the cross sections is necessarily complex and is thankfully confined to an appendix which gives full details leaving the main text to give a clear exposition of the process. It would be useful to have the view of potential users of this technology and how excited they would be to have a new source.

Reviewer #3 (Remarks to the Author):

In the manuscript "Nanophotonics for Pair production", the authors present a novel idea for enhancing the process of pair production, i.e., the generation of electron-positron pairs. Their solution is based on the scattering of gamma radiation off polaritons generated in nanophotonic structures using intense incident light. The usage of light which is not in free space increases the probability of pair production due both to the confinement of light by the nanostructure and the additional momentum components of the trapped light. The authors exemplify their analysis in two examples – surface polaritons and nanogap polaritons. They show that unlike free-space photon scattering, there is a non-zero probability for pair production even in softer gamma rays (MeVs and not GeVs). Part of the impact of this work is the contribution to the relation between the fields of nanophotonics and high-energy physics.

The analysis presented in the manuscript and the supplementary material is extensive and sound, and based on well-established theoretical grounds: both from the field-theory side (pair production using minimal coupling Hamiltonian) and for the nanophotonics side (polaritonic waveforms in relevant experimental platforms). The authors emphasized the applicability of their analysis using numerous numerical examples of simulating standard nanophotonic structures and back-of-the-envelope calculations.

I believe that this work is of significance to the communities of nanophotonics and of high-energy physics. Although the concept of nanophotonic enhancement of light-matter interaction with energetic particles has been investigated in previous works (e.g., Rivera et al. Nature Physics 2019, Kurman et al. PRL 2020), the manuscript presents highly creative original results that go in a completely new direction.

Therefore, I believe that this work should be published in Nature Comm, and suggest below some points that could improve the manuscript.

1. The supplementary material (SM) contains some equations that may be especially valuable to highlight in the main text. Most of the equations in the manuscript constitute known results (Eq. 1, 2a, 2b), and the ones that do constitute new results (Eq. 3) is very general. It would help to provide in the main text some of the concrete equations that are easier to understand, and can be related to the figures. As an example, the difference between plots inside Fig. 3b could be more understandable by citing Eq. S16 or parts of it (at least the prefactors) in the manuscript as an emphasis of the dependence on $R_p, \omega_p, \omega_\gamma$.

A related recommendation: Fig. S2 and especially (a) and (b) provide great graphical demonstrations of distinctive features related to the polariton-assisted pair production, in addition to the existing Fig. 2.

2. It would help to clarify the physical difference between the different diagrams (Fig. 1b). Although it is mentioned that one represents an emission and the other an absorption of a single polariton (and there is a +- summation over them in Eqs. 2,3), I wonder whether one process dominates the other under some conditions (apart from the absence of polariton to absorb), or whether one can be enhanced over the other. In other words, are these two processes different, or is their difference does not have any significance, like diagrams with different spin values of the electron and positron?

3. There is a valuable discussion about the prospects of demonstrating the effect. A few suggestions to further strengthen this discussion:

3.a. Are there setups (probably FELs) that can focus gamma rays inside the gap and thus avoid competing signal from BH? In such cases, the repetition rate of the gamma photons (matched with the repetition rate of the polariton-generating light pulses) will determine the number of events that can be expected per second.

3.b. Seems like a main challenge in attempting to observe the effect is the competing BH signal. Should one aspire for focused gamma photons (possibly increasing damage) or less focused ones (possibly reducing cross section)?

4. A small correction: in the 2nd paragraph (near the citation of ref 17) there seem to be an extra closing parenthesis without an opening one.

Reviewer #1. This paper proposes an interesting mechanism in order to produce electron-positron pairs based on the interaction between the electromagnetic field generated in polaritonic modes and a gamma ray. The calculation of the corresponding interaction cross-section is carried out following standard procedures, and the paper comes with pedagogical appendices on calculation techniques which will certainly prove useful on their own. The main result of this calculation is that the cross-section can be much larger than the Bethe-Heitler cross-section for gap polaritons, thus opening new venues for pair production.

The paper is well written and clearly structured. I have only a few comments.

Our reply. We thank the reviewer for the positive and constructive feedback.

Reviewer #1. 1) In introduction pair production mechanisms and experimental realisations are reviewed. One of the main points being raised is that the $gg \rightarrow e^+e^-$ (Breit-Wheeler) scattering has too low a cross-section to be practical. As a suggestion, I think the paper could benefit from a broadening of the scope of the introduction by mentioning the important astrophysical implications of this process. A general review was done in

<https://ui.adsabs.harvard.edu/abs/2010PhR...487....1R/abstract>. Some more specific examples with suggested references are listed below:

- absorption of TeV gamma rays by the extra-galactic background light

<https://ui.adsabs.harvard.edu/abs/2015ApJ...812...60B/abstract>

- absorption of high-energy gamma rays in the immediate environment of compact sources such as Active Galactic Nuclei (AGN) or during gamma-ray bursts (GRBs)

<https://ui.adsabs.harvard.edu/abs/2019A%26A...627A.159H/abstract> ,

<https://academic.oup.com/mnrasl/article/475/1/L1/4693845>

- plasma production in neutron-star magnetospheres

<https://ui.adsabs.harvard.edu/abs/2018MNRAS.474.1436V/abstract>

Our reply. We thank the reviewer for providing these references, which we have included in the revised introduction to give the paper a broader scope on implications for astrophysics.

Reviewer #1. 2) Not being a specialist of polaritons, I cannot evaluate in detail the relevance of all the statements made concerning these physical objects. Given that the paper is published in a journal targetted at a broad audience, I would recommend to give more references on general polariton physics and in particular the expressions used for their electric field since this is of prime importance for the obtained results.

Our reply. We have strengthened the description of polaritons for a general audience in the revised introduction and also through references for the expressions used for polaritons in the Results section.

Reviewer #2

Reviewer #2. This paper proposes an entirely novel method of producing positrons from the collision of two photons to create an electron/positron pair. The novelty arises in that one of the photons has relative low energy of the order of 1eV in contrast to the second member of the pair which would be a gamma ray. Cross sections for this process would normally be too small to be of use but in this work enhancement is sought by capturing the low energy photons in a polaronic system where the intensity of the polaronic field can be concentrated in a narrow gap. This concentration is a well understood process but to my knowledge has not been exploited in the context of positron production. According to the authors' calculations the method gives several orders of magnitude improvement over existing technology.

Our reply. We thank the reviewer for the positive and constructive feedback.

Reviewer #2. The authors might consider whether a simpler solution would be a rough silver surface to enhance the polaronic fields, as in giant Raman scattering.

Our reply. This is a good suggestion, although BH emission from the bulk of the material would overshadow the proposed new mechanism. This is why we believe that having the emission from vacuum, as in the proposed gap geometry, is a more plausible configuration. We have included this discussion in the revised conclusions.

Reviewer #2. Derivation of the cross sections is necessarily complex and is thankfully confined to an appendix which gives full details leaving the main text to give a clear exposition of the process. It would be useful to have the view of

potential users of this technology and how excited they would be to have a new source.

Our reply. We hope the paper will have good visibility, so potential users will see it and hopefully be excited by this new mechanism. Practical uses could include a direct way to generate ultrashort positron pulses originating from a nanoscale spatial region, as we mention in the revised conclusions and abstract.

Reviewer #3

Reviewer #3. In the manuscript “Nanophotonics for Pair production”, the authors present a novel idea for enhancing the process of pair production, i.e., the generation of electron-positron pairs. Their solution is based on the scattering of gamma radiation off polaritons generated in nanophotonic structures using intense incident light. The usage of light which is not in free space increases the probability of pair production due both to the confinement of light by the nanostructure and the additional momentum components of the trapped light. The authors exemplify their analysis in two examples – surface polaritons and nanogap polaritons. They show that unlike free-space photon scattering, there is a nonzero probability for pair production even in softer gamma rays (MeVs and not GeVs). Part of the impact of this work is the contribution to the relation between the fields of nanophotonics and high-energy physics.

The analysis presented in the manuscript and the supplementary material is extensive and sound, and based on well-established theoretical grounds: both from the field-theory side (pair production using minimal coupling Hamiltonian) and for the nanophotonics side (polaritonic waveforms in relevant experimental platforms). The authors emphasized the applicability of their analysis using numerous numerical examples of simulating standard nanophotonic structures and back-of-the-envelope calculations.

I believe that this work is of significance to the communities of nanophotonics and of high-energy physics. Although the concept of nanophotonic enhancement of light-matter interaction with energetic particles has been investigated in previous works (e.g., Rivera et al. Nature Physics 2019, Kurman et al. PRL 2020), the manuscript presents highly creative original results that go in a completely new direction.

Therefore, I believe that this work should be published in Nature Comm, and suggest below some points that could improve the manuscript.

Our reply. We thank the reviewer for the positive and constructive feedback.

Reviewer #3. 1. The supplementary material (SM) contains some equations that may be especially valuable to highlight in the main text. Most of the equations in the manuscript constitute known results (Eq. 1, 2a, 2b), and the ones that do

constitute new results (Eq. 3) is very general. It would help to provide in the main text some of the concrete equations that are easier to understand, and can be related to the figures. As an example, the difference between plots inside Fig. 3b could be more understandable by citing Eq. S16 or parts of it (at least the prefactors) in the manuscript as an emphasis of the dependence on $R_p, \omega_p, \omega_\gamma$. A related recommendation: Fig. S2 and especially (a) and (b) provide great graphical demonstrations of distinctive features related to the polariton-assisted pair production, in addition to the existing Fig. 2.

Our reply. We thank the reviewer for the recommendations. We have moved Eq. S16 to the main text (new Eq. (4)), as we agree that it delivers a valuable message. However, we prefer to maintain Fig. S2 in the SI: it has tutorial purposes in the present context, but this geometry is not practical to identify the proposed scattering process because the BH mechanism would be dominant; this is in contrast gap polaritons, which is discussed after surface polaritons in the main text.

Reviewer #3. 2. It would help to clarify the physical difference between the different diagrams (Fig. 1b). Although it is mentioned that one represents an emission and the other an absorption of a single polariton (and there is a +- summation over them in Eqs. 2,3), I wonder whether one process dominates the other under some conditions (apart from the absence of polariton to absorb), or whether one can be enhanced over the other. In other words, are these two processes different, or is their difference does not have any significance, like diagrams with different spin values of the electron and positron?

Our reply. We thank the reviewer for raising this point. In fact, Fig. 1b represents four diagrams: two of them related to polariton absorption (those with the orange arrow pointing toward the vertex), and the other two standing for polariton emission (orange arrow away from the vertex). These four diagrams are also organized depending on the order of the interaction (electron and then positron, or the other way around). In addition, we average the results over gamma-ray polarizations and sum over electron and positron spins. These details were not clearly stated in the previous version of the manuscript, so we have improved the explanation on the diagrams of Fig. 1b. In addition, because the polariton energy is so small compared with the pair-production threshold, the processes of polariton emission and absorption contribute with nearly identical amounts to the total cross section. We have also pointed this out in the revised manuscript.

Reviewer #3. 3. There is a valuable discussion about the prospects of demonstrating the effect. A few suggestions to further strengthen this discussion: 3.a. Are there setups (probably FELs) that can focus gamma rays inside the gap and thus avoid competing signal from BH? In such cases, the repetition rate of the gamma photons (matched with the repetition rate of the polariton-generating light pulses) will determine the number of events that can be expected per second.

3.b. Seems like a main challenge in attempting to observe the effect is the competing BH signal. Should one aspire for focused gamma photons (possibly increasing damage) or less focused ones (possibly reducing cross section)?

Our reply. We thank the reviewer for these suggestions. We have mentioned the possibility of using focused gamma rays in the revised conclusions, although that represents a challenge in itself. Also, FELs deliver photons with an energy up to a few 100s keV at most. We thus favor the use of gap plasmons and positron collection from the gap region as a plausible way to demonstrate the proposed pair-production mechanism.

Reviewer #3. 4. A small correction: in the 2nd paragraph (near the citation of ref 17) there seem to be an extra closing parenthesis without an opening one.

Our reply. Thank you for spotting this typo, which is now corrected.

We look forward to hearing from you about the status of our paper.

REVIEWERS' COMMENTS

Reviewer #1 (Remarks to the Author):

The paper has been improved in several aspects accordingly to the recommendations of the first round of reviews. I have no further comments and recommend this paper for publication.

Reviewer #2 (Remarks to the Author):

the revised manuscript meets all concerns raised and I recommend publication

Reviewer #3 (Remarks to the Author):

The manuscript has been improved by the authors in accordance with the previous comments. I recommend publishing without additional modifications.

Reviewer #1. The paper has been improved in several aspects accordingly to the recommendations of the first round of reviews. I have no further comments and recommend this paper for publication.

Reviewer #2. the revised manuscript meets all concerns raised and I recommend publication.

Reviewer #3. The manuscript has been improved by the authors in accordance with the previous comments. I recommend publishing without additional modifications.

We look forward to hearing from you about the status of our paper.